# An Empirical Grid Model for Precipitable Water Vapor

**Xinzhi Wang** [1,2] , **Fayuan Chen** [1,*], **Fuyang Ke** [1,2] and **Chang Xu** [3]

1 School of Remote Sensing and Geomatics Engineering, Nanjing University of Information Science and Technology, Nanjing 210044, China

2 Wuxi Institute, Nanjing University of Information Science and Technology, Wuxi 214100, China

3 Zhejiang Institute of Communications, Hangzhou 311112, China

* Correspondence: 20201248037@nuist.edu.cn

**Abstract:** Atmospheric precipitable water vapor (PWV) is a key variable for weather forecast and climate research. Various techniques (e.g., radiosondes, global navigation satellite system, satellite remote sensing and reanalysis products by data assimilation) can be used to measure (or retrieve) PWV. However, gathering PWV data with high spatial and temporal resolutions remains a challenge. In this study, we propose a new empirical PWV grid model (called ASV-PWV) using the zenith wet delay from the Askne model and improved by the spherical harmonic function and vertical correction. Our method is convenient and enables the user to gain PWV data with only four input parameters (e.g., the longitude and latitude, time, and atmospheric pressure of the desired position). Profiles of 20 radiosonde stations in Qinghai Tibet Plateau, China, along with the latest publicly available C-PWVC2 model are used to validate the local performance. The PWV data from ASV-PWV and C-PWVC2 is generally consistent with radiosonde (the average annual bias is −0.44 mm for ASV-PWV and −1.36 mm for C-PWVC2, the root mean square error (RMSE) is 3.44 mm for ASV-PWV and 2.51 mm for C-PWVC2, respectively). Our ASV-PWV performs better than C-PWVC2 in terms of seasonal characteristics. In general, a sound consistency exists between PWV values of ASV-PWV and the fifth generation of European Centre for Medium-Range Weather Forecasts Atmospheric Reanalysis (ERA5) (total 7381 grid points in 2020). The average annual bias and RMSE are −0.73 mm and 4.28 mm, respectively. ASV-PWV has a similar performance as ERA5 reanalysis products, indicating that ASV-PWV is a potentially alternative option for rapidly gaining PWV.

**Keywords:** ASV-PWV model; radiosonde; ERA5; Qinghai Tibet Plateau

## 1. Introduction

Atmospheric water vapor is mostly stored in the troposphere and accounts for a relatively small proportion, but it has a profound influence on the hydrological cycle, the atmospheric circulation, and the evolution of synoptic systems [1–3]. The precipitable water vapor (PWV) is often used to characterize the water vapor content. Therefore, monitoring PWV is crucial for climate research and extreme weather warning [4–6]. Plenty of techniques (e.g., ground-based photometers, radiosondes, satellite remote sensing, global navigation satellite system (GNSS), and data assimilation) can be used to measure (or retrieve) the PWV. Using ground-based photometers to obtain PWV is simple and inexpensive. Such an approach can offer a higher spatial and temporal resolution than radiosonde data, but less coverage than from satellite measurements, the downside being that it is vulnerable to cloud cover when in use [7,8]. In-situ balloon borne radiosonde is one of the most accurate methods to map the PWV but for local to regional scale with a low spatial and temporal resolution [9,10]. Moreover, radiosonde has a high operating cost and its observations are often disrupted by the artificial shifts caused by equipment changes and the operating procedure [11–13]. Satellite remote sensing using IR emission channels [14] provides an alternative option for retrieving the high temporal and spatial resolution PWV data, but it suffers strongly from the effects from clouds, precipitation, and surface reflection

spectrum uncertainty [15], and PWV values from IR emission channels are available only over the oceans. In many regions, numerous scholars have compared PWV obtained by IR emission channels and GNSS. Kumar Sanjay et al. [16] presented an analysis of GPS data on the campus of Banaras Hindu University, Varanasi and compared the variability of water vapor from Kanpur GPS, AERONET, and MODIS water vapor data. The results indicate that the observation of PWV from GPS is well-correlated with satellite observation (MODIS), and the PWV estimated from GPS data is found to be very sensitive with surface temperature, monsoon onset and rainfall. Shi et al. [17] evaluate the accuracies of the water vapor data MODIS, MERSI, and the visible infrared radiometer (VIRR) using ground-based data (CARE-China and AERONET) as a reference and characterize the local water vapor variations in China. The results indicate that PWV values in China exhibited large spatial and temporal variations (especially in the Qinghai-Tibet Plateau to the South China Sea). The PWV values were obtained from MODIS, VIRR, and MERSI data, and the accuracies varied widely with relative errors from 10% to 899%. Khaniani et al. [18] used one year of ground-based observation from a network of 38 GPS stations to evaluate the performance of MODIS Near Infrared (Near-IR) PWV over Iran. The results indicate that times series between MODIS and GPS both methods agree very well, and the bias values and the height of the stations have a linear relation. Due to the unique superiority (e.g., all-weather, low cost and high accuracy), the GNSS-based PWV detection has achieved dramatic developments in recent years. Unfortunately, the GNSS-derived PWV are often inhomogeneous and have relatively low temporal resolutions [19], resulting from the sparse distribution of GNSS stations. Another powerful method to gather PWV data is the reanalysis data set, which is produced by data assimilation using different atmospheric observations and circulation models [20]. With respect to the sensor-based methods, reanalysis products have advantages of global coverage and spatial integrity, but large uncertainties still exist in the areas where no or limited observations are available for data assimilation [21,22].

Since the distribution in the troposphere is heterogeneous and anisotropic [10], gathering PWV data with high spatial and temporal resolutions confronts the climatology community with a challenge. In recent years, many attempts have been made for this issue. Cortés et al. [23] validate the use of submillimeter tippers to be used at other sites and use the PWV database to detect a potential signature for Chajnantor area climate change over 20 years. Shikhovtsev et al. [24] present the results of a comparison of the PWV estimated from GNSS measurements and the ERA5 reanalysis values corrected for the relief. It is shown that the best correlation between these quantities is observed under clear sky conditions. Ma et al. [25] proposed a dual-scale method for retrieving PWV maps, based on heterogeneous earth data, which can obtain sub-kilometer resolution PWV, while maintaining accuracy. The academic team led by Yao Yibin in Wuhan University, China, focused on the data fusion methods (e.g., the Gaussian Processes method, spherical cap harmonic analysis, Helmert variance component estimation and generalized regression neural network as well) and established some large-scale PWV fusion models for North America by combining different PWV data (e.g., GNSS, the moderate resolution imaging spectroradiometer (MODIS) and the European Centre for Medium-range Weather Forecasts (ECMWF)) (for details, see Yao et al. [26], Zhang et al. [27], Zhang and Yao [28]). The academic team led by Huang Liangke in Guilin University of Technology, China, systematically evaluated the PWV products of MERRA-2 and ERA5 reanalysis in China based on GNSS site and in-situ ground meteorological observations [29], and proposed a ZTD vertical stratification model based on MERRA-2 data considering spatiotemporal factors [30]. In addition, they also analyzed the spatiotemporal characteristics of GNSS-derived PWV by establishing an atmospheric weighted mean temperature model [31,32]. As an alternative, PWV data can be converted by the zenith wet delay (ZWD) obtained from many empirical models. Classic empirical models by Leckner [33] and Kouba [34] have a certain potential for calculating ZWD, but the water vapor decrease factor is usually constant, yielding systematic errors in different seasons. This has given rise to a growing literature on PWV vertical correction modeling. Dousa and Elias [35] propose an improved

ZWD analytical model and its vertical approximation derived from the original profile of meteorological data can be superior to the conventional methods by a factor of 2–3. More recently, Huang et al. [36] proposed a new PWV vertical correction model, considering the time-varying lapse rate and geographical divisions, which can be applied for whole areas in mainland China. No doubt varying model parameters as well as input reanalysis data have great impacts on both of the data fusion methods and PWV vertical correction methods. This issue motivates us to perform further studies in this direction. We follow the concept of the Askne model [37] and establish a new empirical PWV grid model (called ASV-PWV), which is improved by the spherical harmonic function and vertical correction.

The plan of this paper is as follows. Section 2 describes the fifth generation of ECMWF reanalysis (ERA5) data and field Radiosonde observations included in this study. Particular attention is paid to the empirical ZWD deriving and its improvement by the spherical harmonic function and vertical correction. Section 3 discusses the model evaluation. Both the local and area performances are examined. Some concluding remarks are made in Section 4.

## 2. Materials and Methods

### 2.1. ERA5 Reanalysis Products

ERA5 is the latest ECMWF atmospheric reanalysis product with the highest temporal and spatial resolution, which can provide global atmospheric, land, and ocean parameters [38]. ERA5 replaces the ERA-Interim reanalysis, which began in 2006 [39]. ERA5 uses the latest integrated forecasting systems (IFS) to increase ERA5's horizontal resolution to 31 km (ERA-Interim's horizontal resolution is 80 km) and its time resolution from 6 h to 1 h. As a result, ERA5 captures atmospheric phenomena in greater detail than previous, lower-resolution global reanalyses. A large number of assimilated datasets have been reprocessed to improve ERA5 accuracy, especially in the troposphere. Compared with ERA-Interim, another advantage of ERA5 is that its delay time is 5 days, not 2–3 months [40–42]. The products used in this study include meteorological data (37 atmospheric pressure layers from 1000 hPa to 1 hPa). We focus on the 7381 grid points from 75°E to 105°E and 25°N to 40°N.

### 2.2. Radiosonde Profiles

The Integrated Global Radiosonde Archive (IGRA) is a collection of historical and near real time global radiosonde observations [43]. Sounding-derived parameters are recorded according to separated station files and continue to be updated daily. PWV is one of the derived parameters that can be calculated when the pressure, temperature, and dew point depression are available from the surface to a level of 500 hPa [44].

Meteorological profiles of 20 radiosonde stations (see Figure 1) in Qinghai Tibet Plateau, China, are used for model validation. The radiosonde data are obtained from the Integrated Global Radiosonde Archive Version 2 (IGRA2) [45], which is maintained by the National Climate Data Center (NCDC) of the United States (ftp://ftp.ncdc.noaa.gov/pub/data/igra (accessed on 1 June 2022)). In order to improve the data quality, Radiosonde profiles with less than ten pressure levels of valid data were excluded. The PWV data from ERA5 and radiosonde can be calculated from an integral of the meteorological data as follows [46,47]:

$$PWV = \frac{\sum\limits_{i=2}^{n} (P_i - P_{i-1})(q_i + q_{i-1})}{2g} \tag{1}$$

where $g$ is gravitational acceleration(m/s$^2$), $n$ is the number of atmospheric pressure layers; $P_i$ and $q_i$ are the pressure and the humidity at $i$th atmospheric layer, respectively.

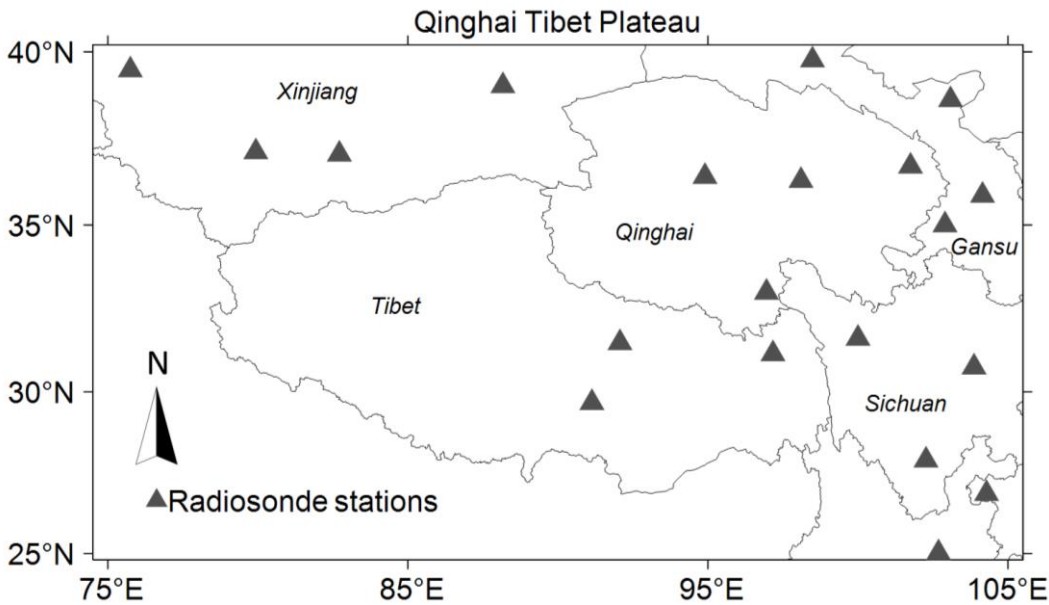

**Figure 1.** Distribution of the 20 radiosonde stations.

*2.3. ASV-PWV Deriving and Correction*

(1)    Fundamentals of ASV-PWV model

Following the model of Askne and Nordius [37], we calculated the approximate ZWD by using the following formula:

$$ZWD = 10^{-6}(k_2' + k_3/T_m)\frac{R_d}{(\lambda + 1)g_m}e_s \tag{2}$$

where $k_2'$ and $k_3$ represent empirically determined refractivity constants, here $k_2' = 22.1 \pm 2.2$ K/hPa, $k_3 = 3.739 \times 10^5 \pm 0.012 \times 10^5$ K$^2$/hPa; $T_m$ is the mean temperature weighted with water vapor pressure (K); $\lambda$ is the water vapor decrease factor; $g_m$ is the mean gravity which equals 9.80665 m/s$^2$; $R_d$ is specific gas constant for dry air which equals 287.0464 J/K/kg; and $e_s$ is the water vapor pressure (hPa) which can be calculated by

$$e_s = \frac{q}{0.378q + 0.622}P \tag{3}$$

where $q$ is specific humidity(kg/kg) and $P$ is atmospheric pressure(hPa).

Then, *PWV* can be converted from *ZWD* by

$$PWV = ZWD \times II \tag{4}$$

where *II* is the water vapor conversion coefficient, which can be calculated by

$$II = \frac{10^6}{(k_3 \times T_m^{-1} + k_2')R_v} \tag{5}$$

where $R_v$ is the gas constant of water vapor(J/K/kg).

Substituting (4) and (5) into (2), we get

$$PWV = \frac{R_d}{(\lambda + 1) \cdot g_m \cdot R_v} \cdot \frac{q}{0.378q + 0.622}P \tag{6}$$

We find that the preliminary ASV-PWV is built up on the three basic parameters *p*, *q*, and $\lambda$. There is evidence that some of these parameters exhibit seasonal characteristics (see Figure 2). In Equation (6), the humidity at a specific atmospheric pressure of 1000 hPa,

*PWV* is obtained by using 5 years (from 2015 to 2019, 1825 days in total) ERA5 atmospheric meteorological data according to Equation (1).

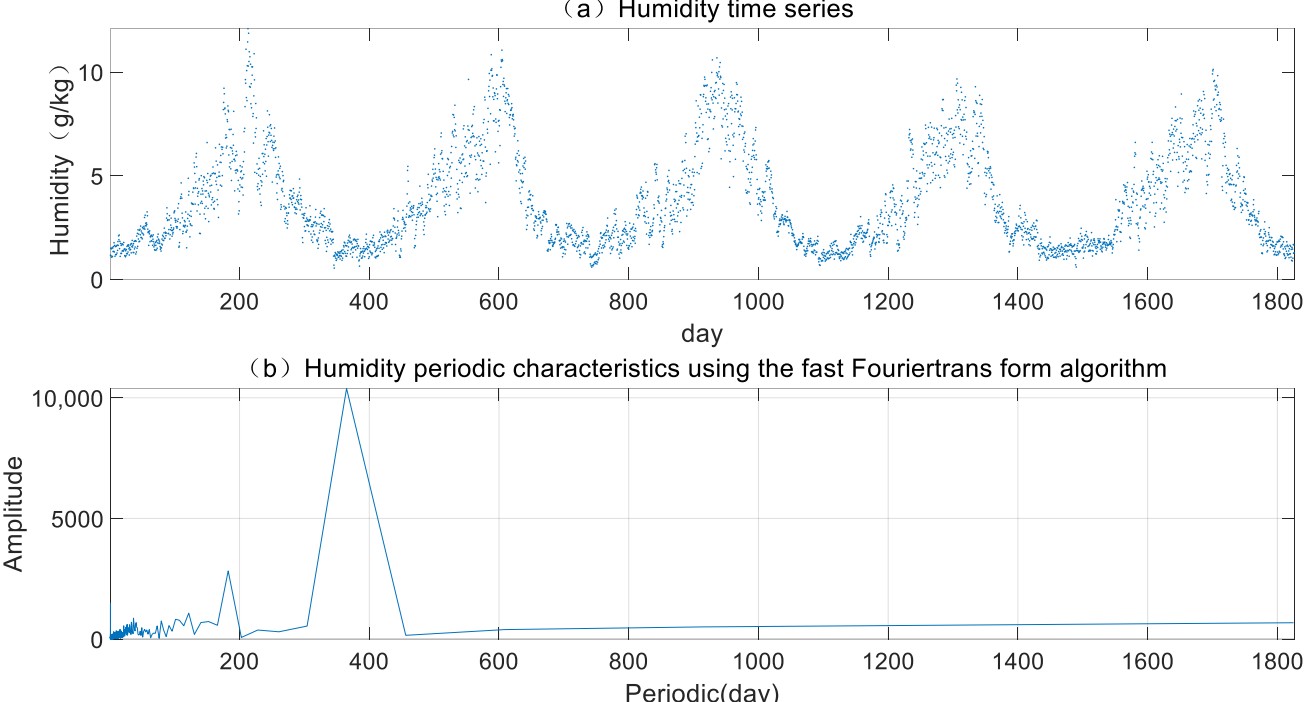

**Figure 2.** (**a**) humidity time series at a specific atmospheric pressure of 1000 hPa and (**b**) its periodic characteristics.

Using the *PWV* outputs from (1), we can obtain the water vapor decrease factor $\lambda$ by (6). The results are shown in Figure 3. Given their significant seasonal nature, we adopt the following seasonal fit formula [48] to deduce empirical temporal information for the coefficients $q$ and $\lambda$. For $q$, it appears as:

$$q = A_0 + A_1 \sin(2\pi \frac{JD}{365.25}) + A_2 \cos(2\pi \frac{JD}{365.25}) + A_3 \sin(4\pi \frac{JD}{365.25}) + A_4 \cos(4\pi \frac{JD}{365.25}) \tag{7}$$

where $A_0$ is intercept, $A_1$ and $A_2$ are annual amplitudes, $A_3$ and $A_4$ are semi-annual amplitudes of the coefficient, and $JD$ is Julian day. Least-squares adjustments are used to fit these parameters to the 7381 grid points that we are interested.

(2)　PWV correction using spherical harmonic function

From the foregoing analysis, we find that both Equations (1) and (6) can be used to determine the PWV data. Assuming $d_{PWV}$ is the difference between the empirical PWV and reanalysis products based PWV, it can be fitted by the spherical harmonic function [49]

$$d_{PWV} = \sum_{n=0}^{N} \sum_{m=0}^{M} (A_{nm} \cdot a_{nm} + B_{nm} \cdot b_{nm}) \tag{8}$$

where $N$ and $M$ are the maximum degree and order of spherical harmonic function. Following Zhao et al. [50], we set the degree to $N = M = 12$ in this study. $A_{nm}$ and $B_{nm}$ are spherical harmonic coefficients, which can be calculated by

$$a_{nm} = P_{nm}(\sin(\varphi)) \cdot \cos(m\gamma) \tag{9}$$

$$b_{nm} = P_{nm}(\sin(\varphi)) \cdot \sin(m\gamma) \tag{10}$$

where $\gamma$ and $\varphi$ are longitude and latitude of the grid point; $P_{nm}(t)$ is the Legendre function with the form

$$P_{nm}(t) = \frac{1}{2^n}(1-t^2)^{\frac{m}{2}} \sum_{k=0}^{\frac{n-m}{2}} (-1)^k \frac{(2n-2k)!}{k!(n-k)!(n-m-2k)!} t^{n-m-k} \tag{11}$$

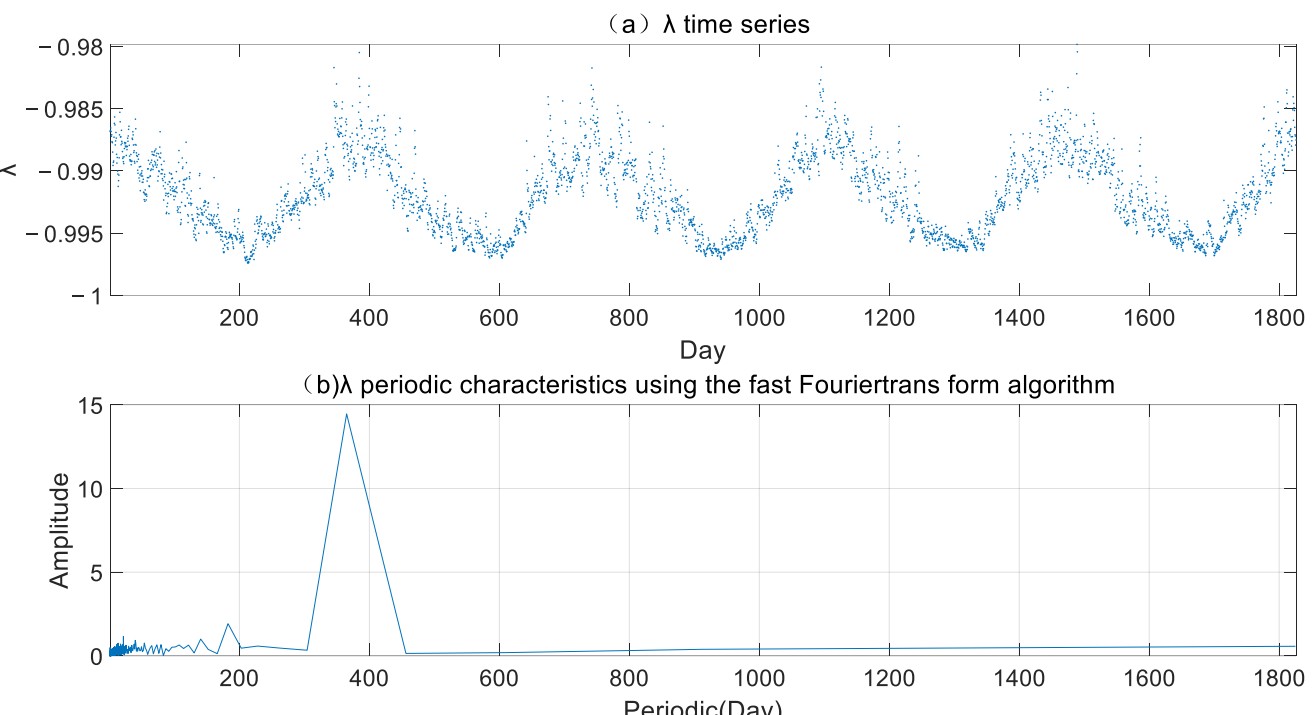

**Figure 3.** Same as Figure 2 but for the water vapor decrease factor.

(3)    Vertical correction

A PWV difference occurs when the elevation difference between the reanalysis product point and the desired position is large [51]. In this study, we selected the ERA5 atmospheric layered meteorological data from 2015 to 2019 to calculate the PWV of different atmospheric pressure altitude and its corresponding atmospheric pressure according to Equation (1). Then, regression analysis is used to analyze the relationship between PWV and pressure. The results are shown in Figure 4.

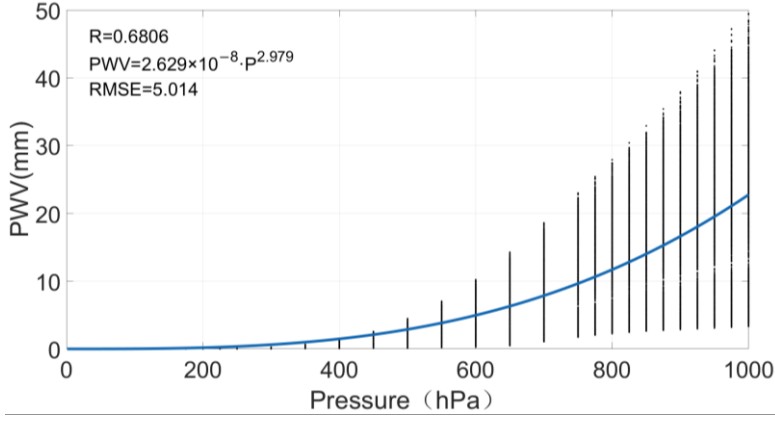

**Figure 4.** Variation relation between PWV and pressure.

Figure 4 shows that there is a significant nonlinear relationship between pressure and *PWV*, which can be expressed by

$$PWV = a \cdot P^b \tag{12}$$

where *a* is the coefficient and *b* is the increase coefficient of *PWV*.

Assuming there are two different atmospheric pressure altitudes P1 and P2 (P2 > P1), we obtain:

$$\frac{PWV_{P_2}}{PWV_{P_1}} = \frac{a \cdot P_2^b}{a \cdot P_1^b} = \left(\frac{P_2}{P_1}\right)^b \tag{13}$$

$$PWV_{P_2} = PWV_{P_1} \cdot \left(\frac{P_2}{P_1}\right)^b \tag{14}$$

where $PWV_{P_1}$ and $PWV_{P_2}$ are respectively the *PWV* of different atmospheric pressure altitudes $P_1$ and $P_2$.

Similar to the parameters *q* and *λ*, the parameter *b* changes with season (see Figure 5). As such, the empirical temporal information for the parameters *b* can again be deduced by using the seasonal fit Equation (7).

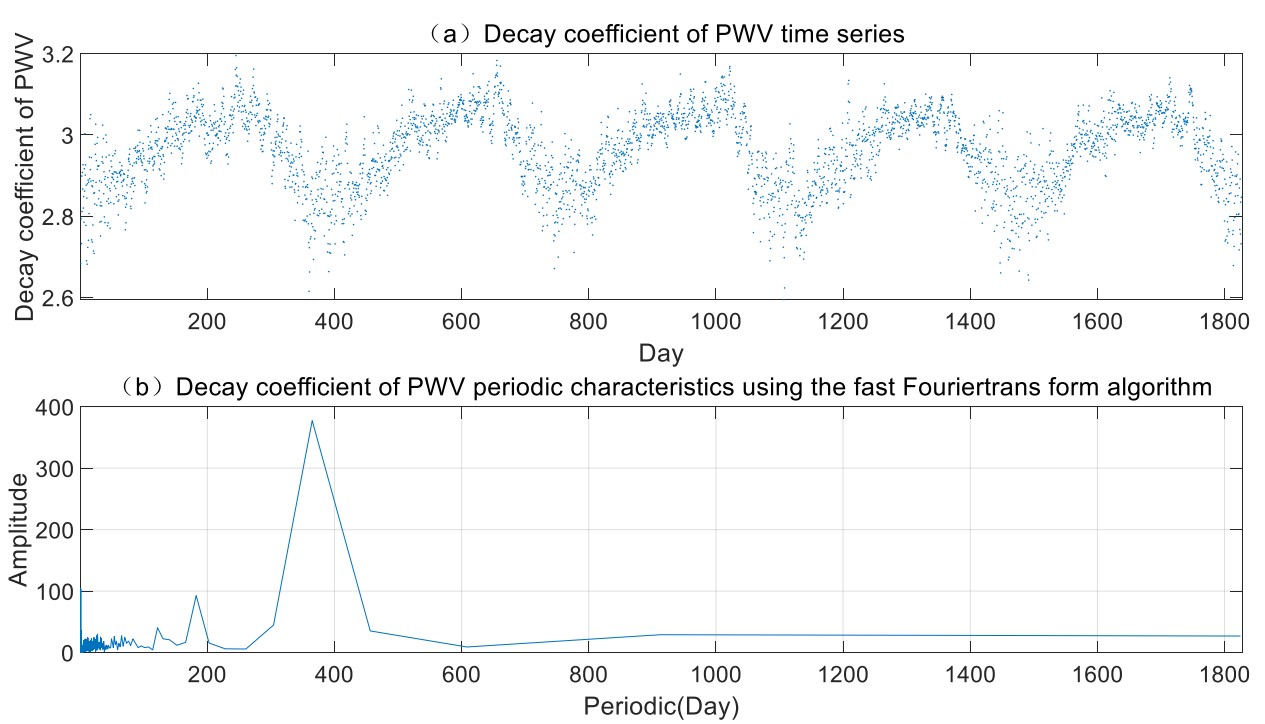

**Figure 5.** Same as Figures 2 and 3 but for the decay coefficient of *PWV*.

Finally, the main expressions of ASV-PWV are Equations (15) and (16):

$$PWV = \frac{R_d}{(\lambda + 1) \cdot g_m \cdot R_v} \cdot \frac{1000q}{0.378q + 0.622} + dPWV \tag{15}$$

$$PWV_{P_2} = PWV_{P_1} \cdot \left(\frac{P_2}{P_1}\right)^b \ (P_2 > P_1) \tag{16}$$

As mentioned already at an earlier stage, the coefficients of *q*, *λ*, *b*, and their amplitudes can be calculated by least-squares based seasonal fitting. For convenience, we store these results in a grid, from which the user could then spatially interpolate the desired position. As a result, ASV-PWV is an alternative option for rapidly gaining PWV with only four parameters of the desired position (e.g., the longitude and latitude, time and atmospheric pressure). Additionally, ASV-PWV has the same spatial resolution as the reanalysis products adopted. For example, if the ERA5 reanalysis data are adopted, ASV-PWV has a high spatial

resolution of $0.25° \times 0.25°$. The next section follows the performance evaluation of our ASV-PWV model.

## 3. Model Validation of ASV-PWV

### 3.1. Statistical Indicators

Radiosonde profiles and ERA5 reanalysis PWV products are used to evaluate the PWV performance of ASV-PWV model. Statistical indicators to measure the performance of ASV-PWV include bias (perfect value = 0) and root mean square error (RMSE, perfect value = 0). Two calculations of indicators are presented as follows:

$$Bias = \sum_{i=1}^{N} \frac{(PWV_{mi} - PWV_{pi})}{N} \tag{17}$$

$$RMSE = \sqrt{\sum_{i=1}^{N} \frac{(PWV_{mi} - PWV_{pi})^2}{N}} \tag{18}$$

where $N$ is the total number of matchups, $PWV_{mi}$ is the PWV of model, and $PWV_{pi}$ is the PWV of Radiosonde profiles or ERA5 reanalysis products.

### 3.2. Validation with Radiosonde Profiles

Profiles of 20 radiosonde stations in Qinghai Tibet Plateau, China, along with the latest publicly-available C-PWVC2 (Huang et al., 2021 [36]) are used to validate the local performance of our model. The results of C-PWVC2 are obtained using the vertical corrected ERA5 reanalysis PWV products (2012–2017), the expressions of C-PWVC2 are Equations (19) and (20):

$$PWV_{h_1} = PWV_{h_2} \cdot \exp(\beta(h_1 - h_2)) \tag{19}$$

$$\beta(\text{DOY}) = -0.453 - 0.037 \sin(2\pi \tfrac{\text{DOY}}{365.25}) - 0.087 \cos(2\pi \tfrac{\text{DOY}}{365.25}) + \\ 0.036 \sin(4\pi \tfrac{\text{DOY}}{365.25}) + 0.023 \cos(4\pi \tfrac{\text{DOY}}{365.25}) \tag{20}$$

where $PWV_{h_1}$ and $PWV_{h_2}$ are respectively the PWV of different height $h_1$ and $h_2$, $\beta$ is decreasing coefficient, DOY is day of year. For more details, see Huang et al. (2021 [36]).

Table 1 and Figure 6 give the annual bias and root mean square error (RMSE) of the ASV-PWV and C-PWVC2 in 2020. With respect to radiosonde data, the annual bias of ASV-PWV is in the range from −2.30 mm to 0.79 mm with a mean of −0.44 mm and the annual bias of C-PWVC2 is in the range from −2.68 mm to −0.32 mm with a mean of −1.36 mm, respectively. The annual RMSE of ASV-PWV is in the range from 1.54 mm to 7.79 mm with a mean of 3.44 mm and the annual RMSE of C-PWVC2 is in the range from 1.42 mm to 4.58 mm with a mean of 2.51 mm, respectively.

Figure 6 shows that the bias of ASV-PWV is mainly distributed around 0 mm, with the range from −1 mm to 1 mm, whereas the bias of C-PWVC2 are all negative and mainly distributed around −1.5 mm. Particularly for the radiosonde in latitude greater than 35° or in longitude greater than 100°, ASV-PWV has larger RMSE than C-PWVC2 accompanied by some abnormally large value, whereas C-PWVC2 is relatively uniform in terms of RMSE and mainly smaller than 3 mm.

**Table 1.** Average annual bias and RMSE of the ASV-PWV and C-PWVC2 in 2020.

| Model | Bias (mm) | | | RMSE (mm) | | |
|---|---|---|---|---|---|---|
| | Avg | Max | Min | Avg | Max | Min |
| ASV-PWV | −0.44 | 0.79 | −2.30 | 3.44 | 7.79 | 1.54 |
| C-PWVC2 | −1.36 | −0.32 | −2.68 | 2.51 | 4.58 | 1.42 |

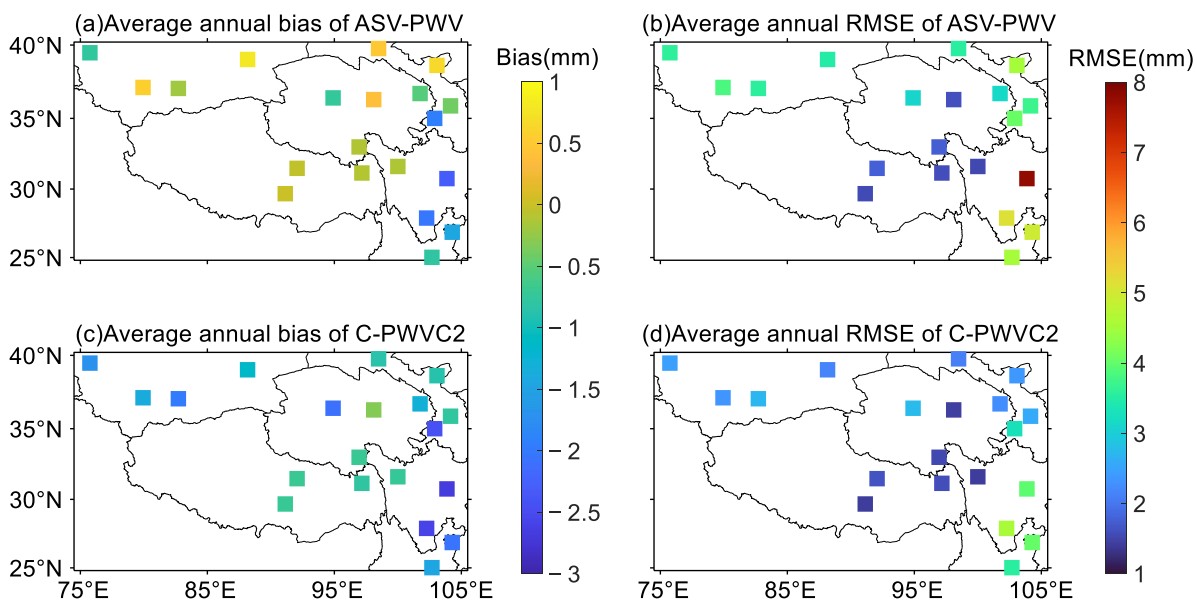

**Figure 6.** Average annual bias and RMSE of the ASV-PWV and C-PWVC2 in 2020.

Table 2, Figures 7 and 8 give the statistical bias and RMSE of ASV-PWV and C-PWVC2 models in four seasons of the year. We find that ASV-PWV has the minimum average bias (0.1 mm) and average RMSE (1.81 mm) in spring and has the maximum average bias (−1.34 mm) and average RMSE (4.83 mm) in autumn, respectively. C-PWVC2 has the minimum average bias (−0.99 mm) in autumn and has the maximum average bias (−1.81 mm) in spring. C-PWVC2 has the minimum average RMSE (1.39 mm) in spring and has the maximum average RMSE (3.31 mm) in summer.

**Table 2.** Performance statistics of the ASV-PWV and C-PWVC2 in 4 seasons of 2020.

| Model | Indicator (mm) | Spring | | | Summer | | | Autumn | | | Winner | | |
|---|---|---|---|---|---|---|---|---|---|---|---|---|---|
| | | Avg | Max | Min | Avg | Max | Min | Avg | Max | Min | Avg | Max | Min |
| ASV-PWV | Bias | 0.10 | 0.95 | −2.86 | 0.14 | 2.71 | −1.61 | −1.34 | 1.34 | −5.37 | −0.65 | 0.50 | −3.49 |
| | RMSE | 1.81 | 4.73 | 0.72 | 3.55 | 8.35 | 1.43 | 4.83 | 10.89 | 2.10 | 2.81 | 6.68 | 0.83 |
| C-PWVC2 | Bias | −1.01 | −0.34 | −1.88 | −1.81 | −0.42 | −3.72 | −0.99 | 0.67 | −2.81 | −1.76 | −0.53 | −5.40 |
| | RMSE | 1.39 | 2.77 | 0.48 | 2.64 | 4.69 | 1.20 | 3.13 | 4.66 | 1.68 | 2.51 | 7.58 | 0.70 |

For both ASV-PWV and C-PWVC2, their bias distribution in the four seasons is generally consistent with that of the entire year (see Figure 7). Some abnormal bias with large values are found in the individual radiosonde located in the southeast region and in autumn and winter. No significant difference of the bias distribution is found in C-PWVC2 in four seasons. Figure 8 shows that the RMSE of ASV-PWV is uniformly distributed in spring and winter but increases in summer and autumn with some larger RMSE values in the radiosonde with latitude greater than 35° or longitude greater than 100°. Meanwhile, the RMSE of C-PWVC2 is uniformly distributed in four seasons. Overall, the ASV-PWV performs better in terms of performance but with lower stability than C-PWVC2. For ASV-PWV, the performance in spring and winter is better than that in summer and autumn, whereas no obvious seasonal changes are found for C-PWVC2. These results reveal that our ASV-PWV performs better than C-PWVC2 in terms of seasonal characteristics.

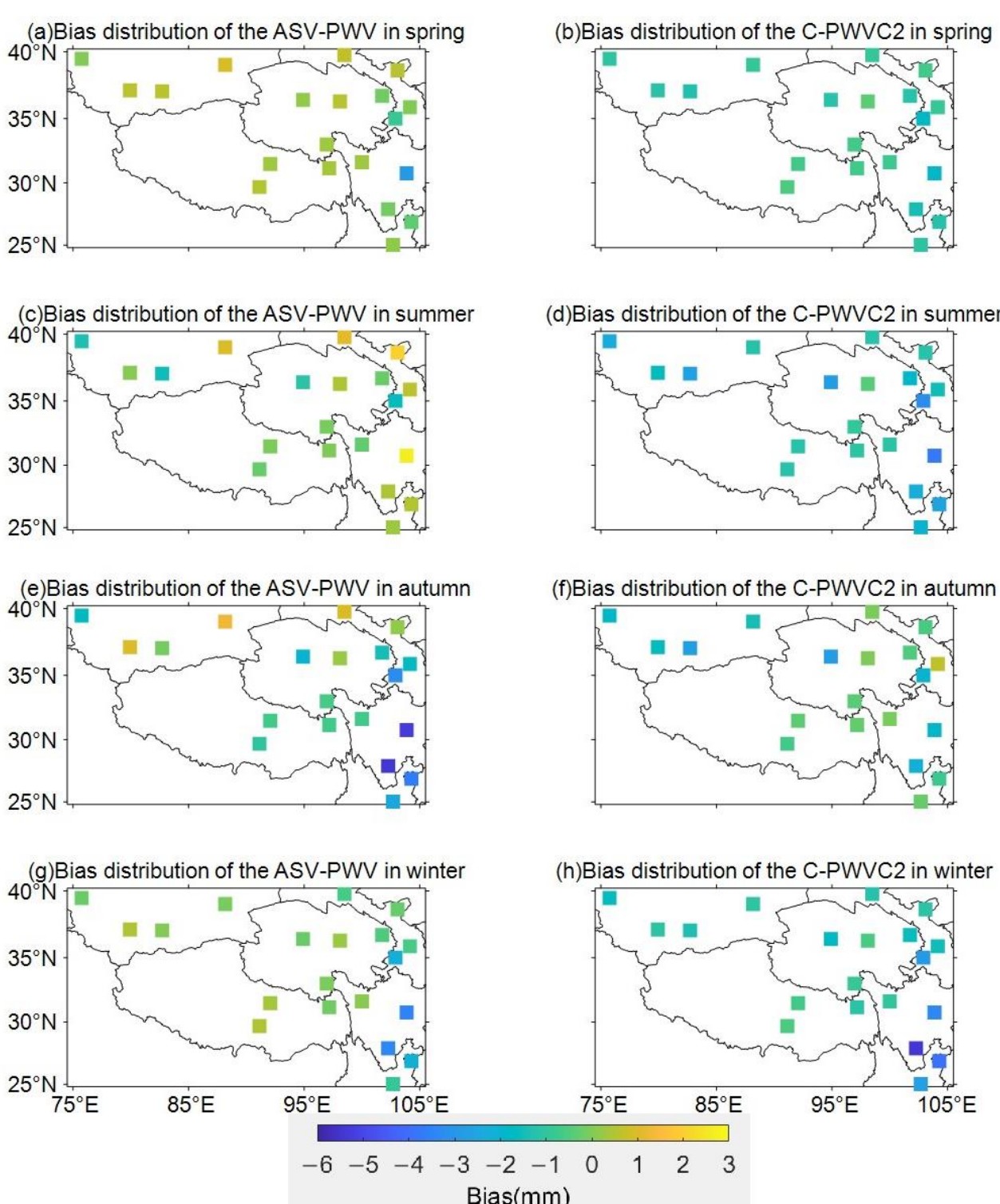

**Figure 7.** Bias distribution of the ASV-PWV and C-PWVC2 in 4 seasons of 2020.

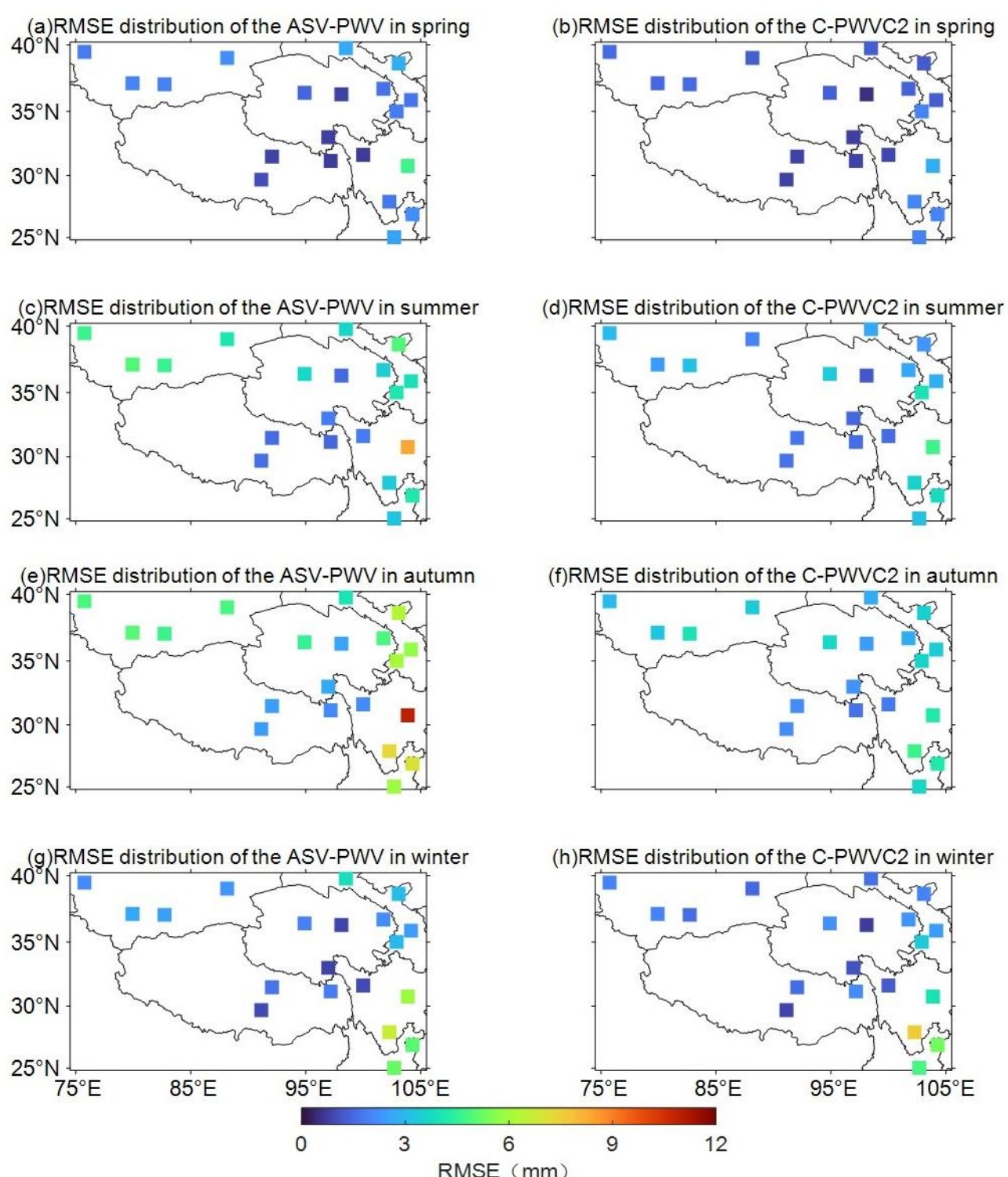

**Figure 8.** RMSE distribution of the ASV-PWV and C-PWVC2 in 4 seasons of 2020.

### 3.3. Validation with the ERA5 Reanalysis PWV Products

In order to further understand the spatial features of our ASV-PWV model in Qinghai Tibet Plateau, a comparison with PWV data from ERA5 is included in this study. Hence, 7381 grid points in 2020 of ERA5 reanalysis products are adopted. Figure 9 shows that the average annual bias and RMSE of ASV-PWV are −0.73 mm and 4.28 mm respectively. Obvious regional characteristics is also found in bias and RMSE distribution (see Figure 9). The bias in the northwest region is lower than that in the southeast region, where the bias is

mainly negative, indicating that the PWV of ASV-PWV is smaller than the ERA5 reanalysis PWV products. Additionally, the RMSE in the central region is obviously better than that in other regions, and large RMSE values are found in southern region. Figures 10 and 11 provide the detail variation of bias and RMSE of ASV-PWV. We find that 90.59% regions have bias in the range of −2.5–0.5 mm, only 2.91% regions have bias less than −2.5 mm, and 6.50% regions have the bias in the range of 0.5–1.5 mm (see Figure 10). We also find that 78.30% regions have RMSE in the range of 1–6 mm, 13.14% regions have RMSE in the range of 6–9 mm, 6.03% regions have RMSE in the range of 9–12 mm, and only 2.22% regions have RMSE larger than 12 mm (see Figure 11).

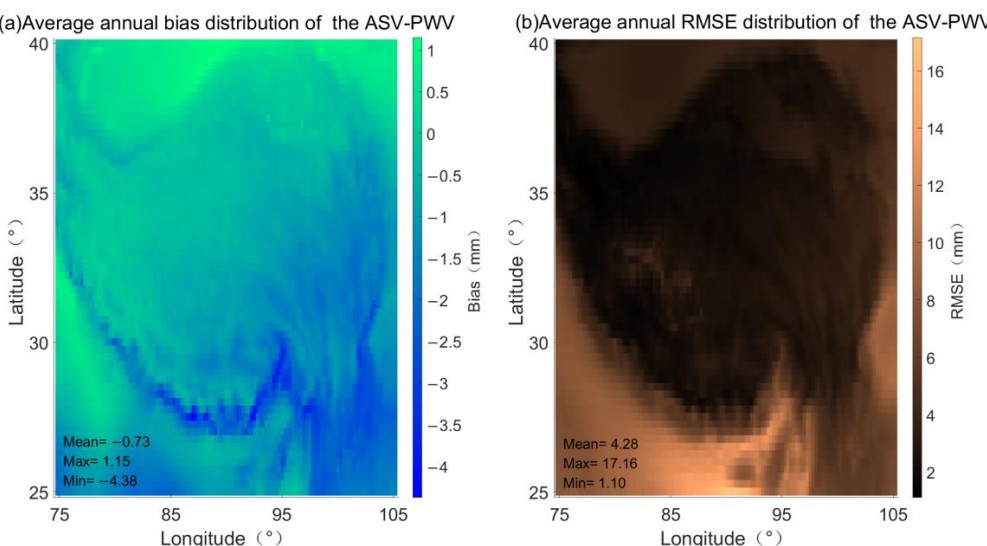

**Figure 9.** Average annual bias and RMSE distribution of the ASV-PWV (with respect to the ERA5 reanalysis PWV products).

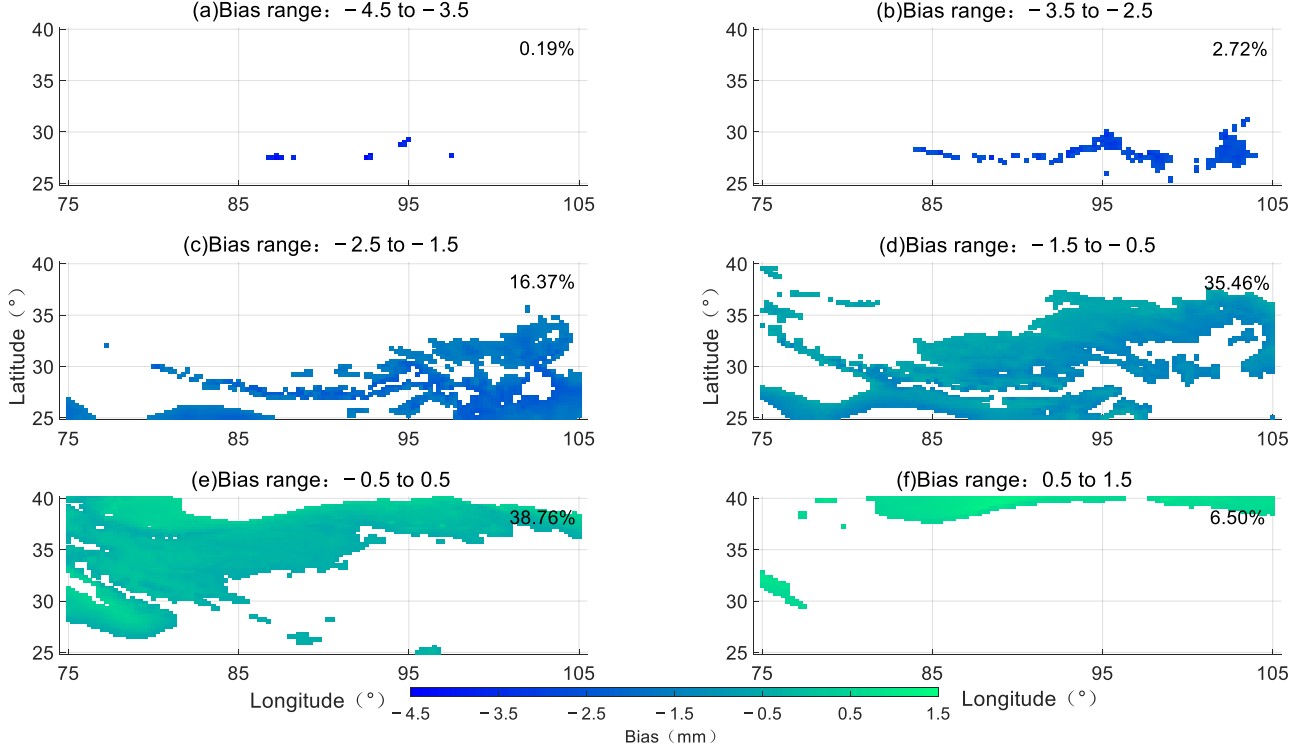

**Figure 10.** The bias distribution of ASV-PWV.

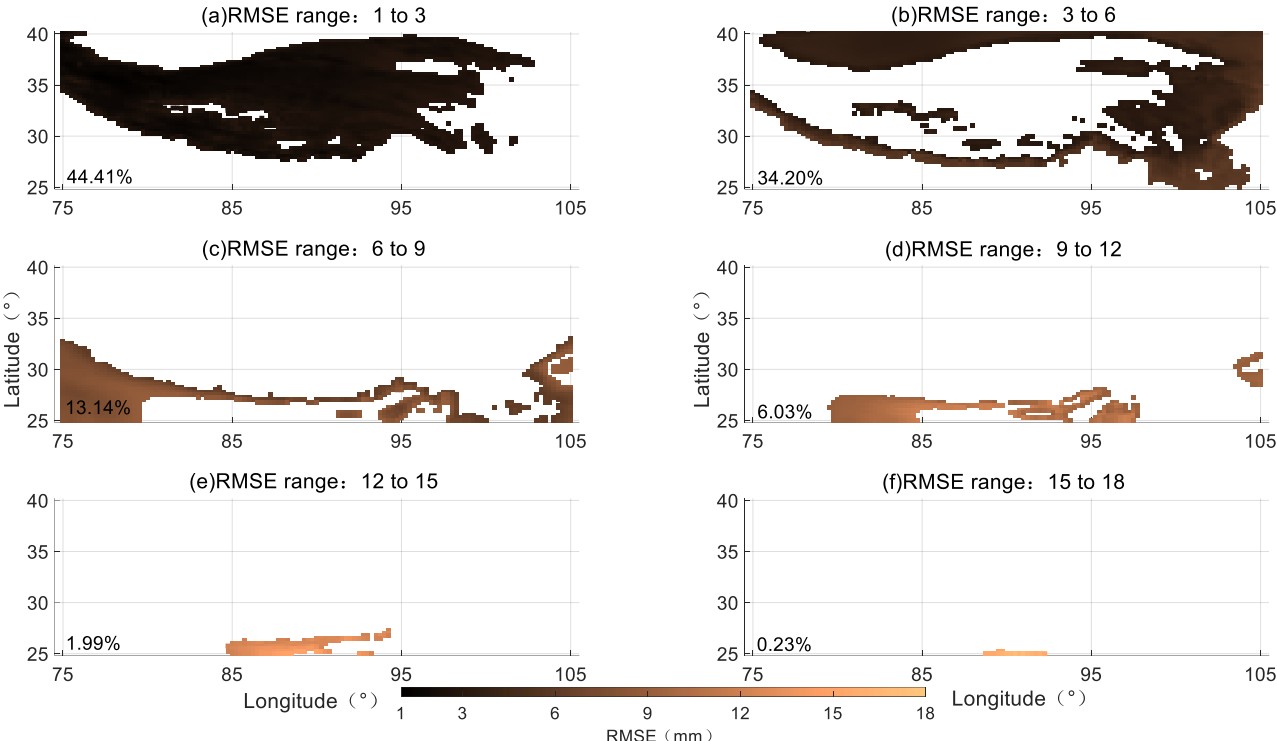

**Figure 11.** The RMSE distribution of ASV-PWV.

Since altitude has a profound influence on PWV as aforementioned, we divide the annual average atmospheric pressure of grid points into different altitudes, to better evaluate the performance of our method (see Figure 12). We find that the bias of ASV-PWV is mainly negative, indicating that the ASV-PWV derived PWV is generally lower than the ERA5 reanalysis PWV products. We also find that the bias generally increases first in the pressure range of 750~800 hPa and then decreases at 850~900 hPa with an average annual bias of −0.73 mm, while the RMSE increases with the atmospheric pressure with a mean of 4.28 mm. All the results show that the performance of our empirical model is basically equivalent to that of ERA5, indicating that ASV-PWV can be applied as a potentially alternative option for rapidly gaining PWV.

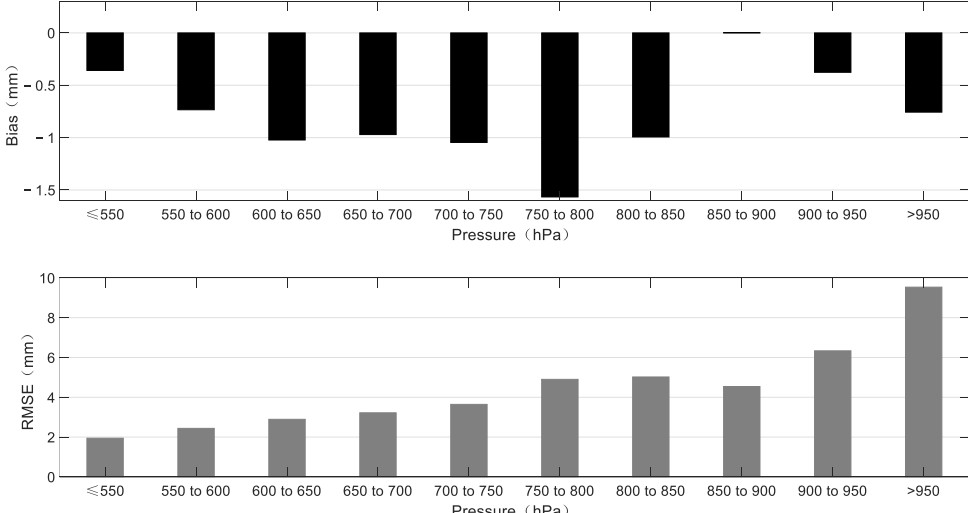

**Figure 12.** Annual average bias and RMSE of ASV-PWV for different atmospheric pressures for ten grid points (with No. of 911, 1774, 798, 500, 441, 403, 409, 731, 279 and 1135 respectively).

## 4. Conclusions

This paper introduces a new empirical PWV grid model (called ASV-PWV) based on the empirical zenith wet delay and improved by the spherical harmonic function and vertical correction. Based on the ERA5 atmospheric meteorological data, some fundamental parameters for empirical PWV model are calculated by least-squares based seasonal fitting and are stored in a grid, from which the user then could spatially interpolate the desired position. As such, ASV-PWV is convenient and enables the user to rapidly gain PWV data with only four input parameters, e.g., the longitude and latitude, time, and atmospheric pressure of the desired position.

Profiles of 20 radiosonde stations in Qinghai Tibet Plateau, China, along with the latest publicly available C-PWVC2 model and PWV data from ERA5 are used to validate the local performance of our model. The PWV value of ASV-PWV and C-PWVC2 models is generally consistent with radiosonde. The results for the whole year show that the annual bias of ASV-PWV is in the range from −2.30 mm to 0.79 mm with a mean of −0.44 mm and the annual bias of C-PWVC2 is in the range from −2.68 mm to −0.32 mm with a mean of −1.36 mm, respectively; the annual RMSE of ASV-PWV is in the range from 1.54 mm to 7.79 mm with a mean of 3.44 mm, and the annual RMSE of C-PWVC2 is in the range from 1.42 mm to 4.58 mm with a mean of 2.51 mm, respectively. The statistical results of the distribution of each radiosonde stations show that the bias of ASV-PWV is mainly distributed around 0 mm, with the range from −1 mm to 1 mm, whereas the bias of C-PWVC2 are all negative and mainly distributed around −1.5 mm. The performance statistics of ASV-PWV and C-PWVC2 models in different seasons over the Qinghai-Tibet Plateau show that ASV-PWV has the minimum average bias (0.1 mm) and average RMSE (1.81 mm) in spring and the maximum average bias (−1.34 mm) and average RMSE (4.83 mm) in autumn, respectively. C-PWVC2 has the minimum average bias (−0.99 mm) in autumn and has the maximum average bias (−1.81 mm) in spring, C-PWVC2 has the minimum average RMSE (1.39 mm) in spring and has the maximum average RMSE (3.31 mm) in summer. For both ASV-PWV and C-PWVC2, their bias distribution in the four seasons is generally consistent with that of the entire year. The RMSE of ASV-PWV is uniformly distributed in spring and winter but increases in summer and autumn with some larger RMSE values in the radiosonde with latitude greater than 35° or longitude greater than 100°, and the RMSE of C-PWVC2 is uniformly distributed in four seasons.

In general, a sound consistency exists between PWV values of ASV-PWV and ERA5 (total 7381 grid points in 2020); their average annual bias and RMSE are −0.73 mm and 4.28 mm, respectively. Obvious regional characteristics are found in bias and RMSE distribution (the bias in the northwest region is lower than that in the southeast region, where the bias is mainly negative, indicating that the PWV data from ASV-PWV is smaller than the PWV data from ERA5. Additionally, the RMSE in the central region is obviously better than that in other regions, and large RMSE values are found in southern region). The statistical results of range distribution of BIAS and RMSE show that 90.59% regions have bias in the range from −2.5 mm to 0.5 mm, only 2.91% regions have bias less than −2.5 mm, 6.50% regions have bias in the range from 0.5 mm to 1.5 mm, and 78.30% regions have RMSE in the range from 1 mm to 6 mm, 13.14% regions have RMSE in the range from 6 mm to 9 mm, 6.03% regions have RMSE in the range from 9 mm to 12 mm, and only 2.22% regions have RMSE larger than 12 mm. Meanwhile, we divide the annual average atmospheric pressure of grid points into different altitudes to evaluate the performance of our method. The results show that the PWV data from ASV-PWV is generally lower than the PWV data from ERA5. The bias generally increases first in the pressure range of 750~800 hPa and then decreases at 850~900 hPa with an average annual bias of −0.73 mm, while the RMSE increases with the atmospheric pressure with a mean of 4.28 mm. ASV-PWV achieves similar performance as the PWV data from ERA5 and radiosonde data, and performs better than C-PWVC2 in terms of seasonal characteristics, indicating that ASV-PWV is a potentially alternative option for rapidly gaining PWV.

Although our ASV-PWV model focuses on the Qinghai Tibet Plateau, it can also be applied to a larger region. Some critical issues (e.g., regional systematic model errors) are not completely taken into consideration. More quality assessments are desirable in future work to evaluate the possible benefits of the ASV-PWV model.

**Author Contributions:** Conceptualization, X.W. and F.C.; methodology, X.W. and F.C.; software, F.C.; validation, X.W., F.C., F.K. and C.X.; formal analysis, X.W., F.C., F.K. and C.X.; investigation, X.W. and F.C.; resources, X.W., F.C., F.K. and C.X.; data curation, X.W. and F.C.; writing—original draft preparation, X.W. and F.C.; writing—review and editing, F.K. and C.X.; supervision, F.K. and C.X.; project administration, F.K. and F.C. All authors have read and agreed to the published version of the manuscript.

**Funding:** This research was funded by the Key Research and Development Program of Jiangsu Province (Social Development Project) [BE2021622], the Natural Science Foundation of Jiangsu Province [BK20211037], the Higher Education Reform Educational Project of Jiangsu Province [2021JSJG219], the Postgraduate Research & Practice Innovation Program of Jiangsu Province [SJCX21_0373], the Science and Technology Project of Changzhou City [CE20225026], the Science and Technology Development Fund Project of Wuxi city [N20201011] and the APC was funded by the Key Research and Development Program of Jiangsu Province (Social Development Project).

**Acknowledgments:** We are grateful to ECMWF for providing ERA5 products and the NCDC for providing IGRA2 products.

**Conflicts of Interest:** The authors declare no conflict of interest.

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
