# Peer review of "An Empirical Grid Model for Precipitable Water Vapor"

_remotesensing, doi:10.3390/rs14236174_

Round 1

Reviewer 1 Report (Previous Reviewer 2)

The author's answer from the attachment can be accepted by the Editor of the Remote Sensing magazine, which could resolve this situation in favor of the author. Still, in that case, it would be one more paper on the list of scientific papers that do not take into account the errors of the physical quantities they analyze, such as the authors themselves state.

Author Response

Reviewer 2 Report (Previous Reviewer 1)

I think the new manuscript has been significantly improved and now warrants publication in remote sensing.

Author Response

Reviewer 3 Report (New Reviewer)

An empirical grid model for Precipitable Water Vapor

The manuscript presented by the authors is devoted to one of the problems that has not been solved in world science. This problem is related to obtaining accurate estimates of the water vapor content in the atmospheric column. Within this task, the authors propose a method for correcting the values ​​of precipitable  water vapor. We support this method, our attitude is generally positive. However, we can draw attention to some aspects of this manuscript and outline recommendations for improving the content of the manuscript. Let's consider one of our fundamental remarks, which appeared when we reviewed this manuscript. Our remark is related to formula 12: ??? = ? ⋅ ?^b. According to formula 12, PWV is related to atmospheric pressure P. At the same time, the content of precipitable water vapor is not directly related to air pressure. In other words, situations are often observed in nature when, at the same pressure, the PWV values ​​can vary within significant limits. For example, the analysis of radio sounding data shows that the correlation coefficients between ??? and ?^b are not high and amount to 0.4 – 0.6. In this regard, we ask the authors to provide graphs with dependences of ??? on P, and also to estimate the spread of these values. This is the main note to the work.

- In order to improve the manuscript, the introduction of the manuscript can be also extended. A large number of studies have been devoted to the study of water vapor. And we recommend to authors to consider the following papers:

I) F. Cortés et al Twenty years of precipitable water vapor measurements in the Chajnantor area / A&A 640, A126 (2020). In this paper, the authors analyze the ratios of the PWV estimated at the peak to the PWV  at the plateau.

II) Shikhovtsev, A.Y., Khaikin, V.B., Mironov, A.P. et al. Statistical Analysis of the Water Vapor Content in North Caucasus and Crimea. Atmos Ocean Opt 35, 168–175 (2022). The paper presents the results of a comparison of the  PWV estimated from GNSS measurements and the Era-5 reanalysis values ​​corrected for the relief. It is shown that the best correlation between these quantities is observed under clear sky conditions.

III) Ma, X., Yao, Y., Zhang, B., He, C. Retrieval of high spatial resolution precipitable water vapor maps using heterogeneous earth observation data / Remote Sensing of Environment. 2022. 278.113100

The above remarks do not reduce the quality of the work and the idea of ​​taking into account seasonal variations in PWV in the correction of its values.

Round 2

Reviewer 3 Report (New Reviewer)

The authors presented a model for vertical correction of deposited water vapor values using a series of coefficients to be determined for different locations and seasons of the year. We agree with the authors that their approach is valid using pressure as the vertical coordinate. The authors responded to our comments and we recommend the manuscript for publication.

This manuscript is a resubmission of an earlier submission. The following is a list of the peer review reports and author responses from that submission.

Round 1

Reviewer 1 Report

The manuscript proposed a new empirical PWV grid model using the zenith wet delay from the Askne model and improved by the spherical harmonic function and vertical correction, which may provide an alternative option for rapidly gaining PWV data. Some results may be interesting, and I suggest a minor revision. I give my comments below.
    1. More studies about the application of GNSS-derived PWV are worth mentioning.

2. In section 2.3, the equation (5) has three parameters of q,  and P. However, only q and  are seasonally fitted in the paper, but how about the P?

3. In section 2.3, the equation (6) is inconsistent with the description in the paper. The output of equation (6) described in the paper is q, the output of formula (6) is PWV.

4. The paper used the C-PWV2 model as a comparison when using the radiosonde data to validate the local performance of the new empirical PWV grid model. So why the paper did not use the C-PWV2 model as a comparison when using the ERA5 data reanalyze PWV products to validate the local performance of the new empirical PWV grid model.

5. Generally paper is written clearly and has good structure. There are some minor mistakes (this is not complete list, and I put this list to help the Author in improvement of future version of article):

-- P2 L36 change ‘Situ balloonborne Radiosonde’ to ‘Situ balloonborne radiosonde’.

--  P2 L39 change ‘their’ to ‘its’.

--  P2 L43 change ‘suffer’ to ‘ suffers’ .

--  P2 L47 change ‘PWV’ to ‘ PWV data’.

--  P2 L59 change ‘focus’ to ‘focuses’.

   P5 L155 right align the formula.

--  P5 L161 change ‘Regression analysis was’ to ‘regression analysis is’.

--  P5 L162 change ‘The results were’ to ‘The results are’.

--  P6 L164 change ‘Fig.4 variation relation’ to ‘Fig.4 Variation relation’.

   P6 L169-171 change

 ‘Assume that there are two different atmospheric pressure altitudes P1 and P2 , and P2 > P1 . Then, we could get the Equation (13) according to Equation (12). Equation (13) can be further transformed into Equation (14).’

    to

‘Assuming there are two different atmospheric pressure altitudes P1 and P2 (P2 > P1), we obtain:’

P7 L197-L199 modify the formula (14), the formula may be incomplete.

      P8 L217 change ‘ASV-PW’ to ‘ASV-PWV’.

      P8 L233 change ‘Our’ to ‘our’.

P8 Table 2 Title format.

-- P12 L 328-329

change

‘Yao, Y., Xingyu, X. & Yufeng, H., 2018. Establishment of a regional precipitable water vapor model based on the combination of GNSS and ECMWF data, Atmospheric Measurement Techniques Discussions.’

   to

‘Yao, Y., Xingyu, X. & Yufeng, H., 2018.Establishment of a regional precipitable water vapor model based on the combination of GNSS and ECMWF data, Atmospheric Measurement Techniques Discussions. [preprint], https://doi.org/10.5194/amt-2018-227.’

Reviewer 2 Report

The manuscript entitled “An empirical grid model for Precipitable Water Vapor” (remotesensing-1956043-peer-review-v1) discusses the empirical grid model for precipitable water vapor. This manuscript could be interesting to readers because it deals with the important problem of modeling precipitable water vapor. However, the manuscript has disadvantages, which should be corrected in order to become a scientific paper. It will be mentioned here only the most important disadvantages (D) in the manuscript, which should guide the authors in preparing a new revised version of the manuscript.

D1.) Lines 7-24, (Abstract):

D1.1) Quotes should be omitted.

D1.2) MDPI publications use the so-called Vancouver citation system, which is not the case here.

Note 1: This note also applies to the rest of the text.

D1.3) Each abbreviation must be accompanied by its full name.

Note 2: Repetitive terms are usually shortened. If the terms are not repeated, only the full name should be used. This note also applies to the rest of the text.

D1.4) Atmospheric values whose unit is mm, such as precipitation, etc., have an observation error of 1 mm. There must be the same error when these values are calculated in any way. The calculation error cannot be smaller than the measurement/observation error. Also, this means that these sizes cannot have values with decimal places, as is the case here, because that would mean that the error of that size is on the second decimal place, or on one-hundredth of 1 mm, which is very wrong.

Note 3: This note also applies to the rest of the text.

D2.) Line 25, (Keywords): The terms used in the title should be omitted.

D3) Lines 100 and 101, (Figure 1):

D3.1) The units for longitude and latitude are not complete. Four points of the compass are missing.

D3.2) Figure 1 would be clearer if a geographical map of the area with basic attributes was inserted.

D4) Whole manuscript (Mathematical formulas): In mathematical formulas, each variable must be defined and its units must be written.

Reviewer 3 Report

Review of manuscript “An empirical grid model for Precipitable Water Vapor”, remotesensing-1956043-peer-review-v1 submitted to Remote Sensing

General Comments

This is an interesting study about a methodology for producing a grid model for PWV, a relevant topic for the Remote Sensing journal. Overall I’m in favour of accepting the manuscript but some corrections, improvements and clarifications are needed first. Some aspects of the English language need to be improved – I include in the specific comments some of them. Some clarifications are requested regarding the formulas used and some figures should be improved. Due to the number of items listed I suggest a major review.

Specific Comments

1.      Page 1, line 7, suggest: factor -> variable

2.      Page 1, line 12, and all reference citations: note the text references do not follow the journal style, where a super index identifies the reference cited with the numbered list of references ordered by first appearance in section “References”. Please check and correct.

3.      Page 2, 1rst paragraph. I suggest that authors complement the background given by citing here that PWV can also be measured with ground-based photometers citing some references, for example Berezin et al (2017) or Campmany et al (2010).

4.      Page 2, line 31. Suggest: the synoptic system -> synoptic systems (please remove ‘the’ and add ‘s’)

5.      Page 2, line 36. Please check: Situ -> -In situ (typically in italics because it is a Latin term).

6.      Page 5, line 55. Please check if the term ‘remain’ should be deleted – I think the current sentence does not make sense.

7.      Page 2, line 73. Please check the expression ‘no doubting that’ do you mean ‘no doubt’?

8.      Page 3, line 87. Suggest removing ‘products’ to avoid repeating that word in the same sentence.

9.      Page 3, lines 98-99. I cannot understand this sentence: do you mean ‘In order to improve the data quality the data corresponding to the lowest 10 atmospheric pressure layers.’? If that’s the case please comment briefly (one or two sentences) which can be the effect of neglecting the lowest 10 layers as they may contain a substantial moisture content.

10.  Page 3, Figure 1 should be repeated overlying a geographic map so that the reader will better understand it. Similarly in all similar figures such as Figure 6, etc..

11.  Page 3, line 104. Please remove ‘can’.

12.  Page 3, line 107 (and elsewhere starting with a ‘where’, line 113, etc.): please start the first column of the line below the first column of the text (there should be no tabulation).

13.  Page 3, line 110. Suggest: dry constituents -> dry air

14.  Page 4, line 119. Please remove ‘this’.

15.  Page 5, line 148. You mean n and m (lower case) or N and M (upper case)? The summatory indicates that the maximum values are N and M respectively, doesn’t it?

16.  Page 5, line 150: calculate -> calculated

17.  Page 5, line 157. ‘The PWV’ or ‘A PWV’: I’m not sure to which difference do you refer.

18.  Page 5, line 158: suggest select -> selected (verb tense consistency).

19.  Page 5, line 162: suggest were -> are

20.  Page 6, lines 165-166. Please rewrite, cannot be understood.

21.  Page 6, line 168. Please: Where -> where (and below the first column, without tabulation).

22.  Page 6, line 181: I suggest you write equations (15) and (16) just after this sentence, which then should be: “… Equations (15) and (16):”.

23.  Page 7, Equation 14. You use here variable DOY (Day of Year) but in line 135, Equation (7), you use JD (Julian Day). Presumably they are the same so please use only one term for the same concept to avoid possible confusions.

24.  Page 7, line 208. Please correct: 0mm -> 0 mm

25.  Page 8, Figure 6 (and Figure 7). I think the colour scale used for panels (a) and (c) (blueish colours) does not allow to see differences properly. Please look for an alternative colour scale to improve this issue, perhaps with discrete different colours (not continuous) to distinguish at least positive and negative bias.

26.  Page 8, line 233. Typo: Our -> our

27.  Page 8, Table 2. Please add units of Bias and RMS, for example in the second column: Bias (mm) and RMS (mm).

28.  Page 9, line 253. Please use the same number of decimal digits when giving percentages in lines 250 and 253.

29.  Page 12, line 315. Please complete reference, I think something is missing (JGR article code).

30.  Page 12, line 329. Please complete this reference and check if it was accepted.

31.  Page 12, line 341. Please check possible typo ‘147’, shouldn’t it be a day of September?

32.  Page 13, line 343. Please complete reference – page numbering missing.

References

Berezin, I. A., et al. (2017). Error analysis of integrated water vapor measured by СIMEL photometer. Izvestiya, Atmospheric and Oceanic Physics, 53(1), 58-64. https://doi.org/10.1134/S0001433817010030

Campmany, E., et al. (2010). A comparison of total precipitable water measurements from radiosonde and sunphotometers. Atmospheric Research, 97(3), 385-392.  https://doi.org/10.1016/j.atmosres.2010.04.016

Round 2

Reviewer 2 Report

As important disadvantage D1.4 from my first review does not accept the content of the manuscript is meaningless and should be rejected. I repeat:

D1.4) Atmospheric values whose unit is mm, such as precipitation, etc., have an observation error of 1 mm. There must be the same error when these values are calculated in any way. The calculation error cannot be smaller than the measurement/observation error. Also, this means that these sizes cannot have values with decimal places, as is the case here, because that would mean that the error of that size is on the second decimal place, or on one-hundredth of 1 mm, which is very wrong.

Reviewer 3 Report

Dear authors,

Thank you for the new corrected version of the manuscript. I think that now can be accepted for publication.

I simply add a couple of comments so that you can take them into account during the proof/production stage:

1). Line 102. the radiosonde data with the atmospheric pressure layers less than 10 was excluded -> radiosonde profiles with less than ten pressure levels of valid data were excluded

2). Line 131 (and 172, 205) Where -> where [the first letter should be lower case, not upper case].
